# Exploring the Frequency of Homologous Recombination DNA Repair Dysfunction in Multiple Cancer Types

**DOI:** 10.3390/cancers11030354

**Published:** 2019-03-13

**Authors:** Lucy Gentles, Bojidar Goranov, Elizabeth Matheson, Ashleigh Herriott, Angelika Kaufmann, Sally Hall, Asima Mukhopadhyay, Yvette Drew, Nicola J. Curtin, Rachel L O’Donnell

**Affiliations:** 1Northern Institute for Cancer Research, Paul O’Gorman Building, Medical School, Newcastle University, Framlington Place, Newcastle upon Tyne NE2 4HH, UK; lucy.gentles@ncl.ac.uk (L.G.); bojidar@doctors.org.uk (B.G.); lizmatheson1@gmail.com (E.M.); Ashleighjherriott1@gmail.com (A.H.); angelikakaufmann@gmx.net (A.K.); Sally.Hall@newcastle.ac.uk (S.H.); asima7@yahoo.co.in (A.M.); yvette.drew@ncl.ac.uk (Y.D.); 2Northern Centre for Cancer Care, The Newcastle upon Tyne Hospitals NHS Foundation Trust, Freeman Hospital, Newcastle upon Tyne NE7 7DN, UK; 3Northern Gynecological Oncology Centre, Queen Elizabeth Hospital, Sherriff Hill, Gateshead NE9 6SX, UK; 4Tata Medical Center, 14 MAR (E-W), New Town, Rajarhat, Kolkata 700 160, India

**Keywords:** homologous recombination DNA repair, functional biomarker assay, poly(ADP-ribose) polymerase inhibitors (PARPi), primary culture, precision medicine, platinum-based chemotherapy

## Abstract

Dysfunctional homologous recombination DNA repair (HRR), frequently due to BRCA mutations, is a determinant of sensitivity to platinum chemotherapy and poly(ADP-ribose) polymerase inhibitors (PARPi). In cultures of ovarian cancer cells, we have previously shown that HRR function, based upon RAD51 foci quantification, correlated with growth inhibition ex vivo induced by rucaparib (a PARPi) and 12-month survival following platinum chemotherapy. The aim of this study was to determine the feasibility of measuring HRR dysfunction (HRD) in other tumours, in order to estimate the frequency and hence wider potential of PARPi. A total of 24 cultures were established from ascites sampled from 27 patients with colorectal, upper gastrointestinal, pancreatic, hepatobiliary, breast, mesothelioma, and non-epithelial ovarian cancers; 8 were HRD. Cell growth following continuous exposure to 10 μM of rucaparib was lower in HRD cultures compared to HRR-competent (HRC) cultures. Overall survival in the 10 patients who received platinum-based therapy was marginally higher in the 3 with HRD ascites (median overall survival of 17 months, range 10 to 90) compared to the 7 patients with HRC ascites (nine months, range 1 to 55). HRR functional assessment in primary cultures, from several tumour types, revealed that a third are HRD, justifying the further exploration of PARPi therapy in a broader range of tumours.

## 1. Introduction

A major advance in cancer care is the development of precision medicine, which is dependent upon the ability to match anticancer therapies with tumour-specific alterations. Tumour-specific defects in the DNA damage response (DDR) are common across the spectrum of tumour types and are exploitable, provided that they can be reliably identified. This study aims to apply lessons learnt in ovarian cancer to a broader range of cancer types using an unbiased approach. 

The DDR is essential in maintaining cellular viability in the face of the high level of endogenous DNA damage. Dysfunction in a DDR pathway promotes the genomic instability that leads to cancer formation, but also determines the sensitivity to novel therapies targeting a compensatory DDR function by “synthetic lethality” as well as conventional DNA damaging chemotherapy and radiotherapy [1]. The classic demonstration of synthetic lethality is the role of poly(ADP-ribose) polymerase inhibitors (PARPi) in cancers in which homologous recombination DNA repair (HRR) is defective (HRD), particularly in breast and ovarian cancers associated with germline BRCA1 and BRCA2 mutations [2]. There are currently three US Food and Drug Administration/European Medicines Agency (FDA/EMA)-approved PARPi for use in relapsed high grade ovarian cancer: olaparib (Lynparza™; AstraZeneca, Cambridge, UK) [3,4], rucaparib (Rubraca™, Clovis Oncology, San Francisco, CA, USA) [5], and niraparib (Zejula™; Tesaro Inc, Waltham, MA, USA) [6] (see FDA and EMA websites for the most up to date licence indications in this fast-evolving field). PARP inhibitors are now entering first-line ovarian cancer studies as well (see clinicaltrials.gov website). BRCA 1/2 (germline and somatic) mutations are only seen in up to 20% of high-grade serous ovarian cancer [7], but our own group was the first to identify a substantially higher fraction (~55%) of tumours with HRD using a functional assay [8], which was later confirmed with molecular analysis by The Cancer Genome Atlas (TCGA) [9]. Presumably, this explains the progression-free survival (PFS) benefit seen with niraparib [6] in patients without germline BRCA mutations, and more recently with rucaparib in ARIEL2 [5] and ARIEL3 [10]. Various genomic screens are currently being used to identify HRD ovarian cancer without pathogenic BRCA mutation, such as the loss of heterozygosity (LOH) and the mutational analysis of key HRR genes [5]. LOH measures the “genomic scarring” that is likely to persist even when HRR function is restored [11], and mutational analyses will not identify epigenetic or purely phenotypic changes. Functional biomarkers can provide a global measurement of DDR dysfunction in real time without having to identify individual components, many of which are yet to be discovered (reviewed in [12]). Such functional measures are predicted to complement, or even supersede, genomic or transcriptomic analyses, which show poor (20–25%) correlation with protein levels, let alone function [13]. We have established an assay for HRR function in ovarian cancer ascites [8] that predicts ex vivo sensitivity to PARPi and clinical sensitivity to platinum therapy [8,14]. This assay has subsequently been successfully applied to malignant pleural effusions, where we identified a high frequency of HRD in non-small cell lung cancer that was not identified by the next-generation sequencing screening of key HRR genes [15]. Clinical trials are being undertaken to determine the efficacy of PARPi in this tumour type [16,17,18,19,20,21,22,23,24,25,26,27], with data already having been published for some [28,29,30].

Evidence is emerging suggesting that HRR may be important in many other cancers, signifying a possible broader use of PARPi [14]. Pancreatic cancer is of particular interest, with a rising incidence, a disproportionate number of cancer deaths [31,32], and recognition that at least a proportion of pancreatic cancer is associated with germline BRCA mutations [33,34]. 

The aim of the current study was to determine the feasibility of assessing HRR function and PARPi sensitivity in ex vivo primary cultures from patients’ malignant ascites from a variety of cancer types. We believe that this represents the first study of this kind. Ultimately, we aimed to indicate the frequency of HRD in different tumour types to justify further studies in larger patient cohorts with varied malignancies for PARPi therapy, which were stratified by HRR function. 

## 2. Results

### 2.1. Establishing Primary Cultures from Ascitic Fluid Samples

Ascitic fluid samples were collected from 27 consented patients with various cancer types including colorectal, upper gastrointestinal (GI), pancreatic, hepatobiliary (HB), breast, peritoneal mesothelioma, and non-epithelial ovarian cancers. The patient demographics are described in Table 1. The median age was 57 years (range, 26 to 83 years old), with 13/27 patients sampled at primary presentation and the remainder at the time of relapse in the palliative setting. Primary cultures were successfully established for 26/27 samples. Cultures grew as adherent monolayers in distinct confluent clusters, and most of the cells displayed a rounded cobblestone epithelial-type morphology, as shown in Figure 1A. The epithelial origin of primary cell cultures was confirmed morphologically and by pancytokeratin immunofluorescence, as shown in Figure 1B–D. Most (24/26) of the cultures were >90% pancytokeratin-positive, and taken forward for further analysis. The two non-epithelial ovarian cancer cultures were pancytokeratin-negative, but expressed CA125 and vimentin, which was in keeping with their diagnostic tumour immunohistochemistry; thus, they were also included for HRR analysis.

### 2.2. Determining HRR Status in Primary Cultures

The HRR status of primary cultures was successfully determined in 24/26 samples (Figure 2A). Samples PA025 (HB cancer) and PA026 (upper GI cancer) were excluded from analysis due to failure to increase γH2AX foci, indicating that they had failed to replicate during the exposure period to generate collapsed replication forks. Eight cultures (33%), in which there was a >2-fold increase in γH2AX foci, had a <2-fold increase in RAD51 foci (median increase of 0.9 fold: 0.5 to 1.9), and were therefore confirmed as being HRD. These were: 2/8 colorectal, 1/3 upper GI, 2/5 pancreatic, 1/3 HB, 1/2 breast, and 1/1 peritoneal mesothelioma. There was a substantial variation in the increase in RAD51 foci in the HRC cultures following rucaparib exposure, with a median increase of 4.6-fold (2.1 to 13.2), as shown in Figure 2B. 

### 2.3. Growth Inhibition with PARP Inhibitor 

Growth inhibition following exposure to rucaparib was determined in 18 cultures. In the clinical studies of continuous versus discontinuous dosing schedules in ovarian cancer patients, it was evident that continuous treatment was superior [35], but our pre-clinical studies with Capan-1 (BRCA2 mutant pancreatic carcinoma) cells and xenografts suggested that a short exposure to rucaparib was sufficient to produce durable PARP inhibition and antitumour activity [36]. In order to investigate whether continuous exposure to rucaparib was necessary for cytotoxicity in the primary cultures of patients’ ascites cells or whether short exposures would be sufficient, 9 PA cultures were continuously exposed for 10 days, 6 were treated for 48 h, and 3 underwent both continuous and 48-h exposure in parallel. A representative growth inhibition curve response to rucaparib is shown in Figure 3A. As our earlier data showed that 10 μM of rucaparib distinguished between HRR competent (HRC) and HRD in both established cell lines and primary cultures [8,37], we compared the growth of HRD primary cultures with HRC cultures at this concentration. In the 9 fresh samples exposed continuously, mean cell growth was 36.6 ± 8.8% in the HRD samples compared to control untreated cells, and 90.4 ± 7.6% in the HRC samples (Mann–Whitney test, *p* = 0.0238). Including the 3 thawed samples that were exposed continuously in the analysis showed a mean cell growth of 46.5 ± 11.7% in the HRD samples and 75.2 ± 11.5% in the HRC samples (Mann–Whitney test, *p* = 0.2141), Figure 3B (i). There was no difference in the cell growth values of HRD (42.3 ± 10.1%) and HRC (48.2 ± 2.5%) in the 9 thawed cultures exposed for 48 h to rucaparib (Mann–Whitney test, *p* = 0.5556), indicating that continuous exposure was needed to differentiate between HRC and HRD, with the lower growth in cultures exposed for 48 h possibly reflecting a loss of cells when the rucaparib-containing medium was replaced with fresh medium, as shown in Figure 3B (ii).

### 2.4. HRR as a Clinically Prognostic Marker

The clinical data of the study cohort of cancer patients are summarised in Table 1 with all of the treatment regimens determined by the treating oncologist. Thirteen out of 24 of the cultures that were characterised for HRR function were generated from ascites sampled from chemotherapy-naïve patients at the time of primary diagnosis, of which five patients were HRD. The remaining 11 cultures were generated from ascites from patients who had received a median of one line (range 1 to 3) of chemotherapy prior to sampling, of which 3 were HRD. No patients received PARPi. Overall, 10 patients received platinum-based chemotherapy at some point in their treatment, either pre-sampling or post-sampling of ascites. In all 10 patients, this was given as combination therapy. The median overall survival (OS) of the entire cohort was 11 months (one to 90), with only 4/24 (17%) patients alive at the last follow-up. 

The median OS of the patients whose ascites was characterised as HRC (*n* = 16) was 8 months (range: 1 - 58), in comparison to 11 months (range: 2 - 90) in the patients with ascites characterised as HRD (*n* = 8), Figure 4A, (*p* = 0.9). Within the subgroup of 10 patients who had undergone treatment with platinum therapy, the median OS was nine months (range, 4 to 55) in patients whose ascitic culture was characterised as HRC (*n* = 7) in comparison to 17 months (range, 10 to 50) in the patients with ascitic cultures that were characterised as HRD (*n* = 3), as shown in Figure 4B, (*p* = 0.3).

The cases included were diverse in terms of their cancer stage, tumour origin, comorbidities, and most importantly, the level of pre-treatment with chemotherapy prior to sampling for HRR functional assessment. Examples of 2 case studies are given in Figure 5, where the clinical courses of the 2 cases are used for illustration. Figure 5A represents a patient diagnosed with advanced peritoneal mesothelioma who underwent initial treatment with platinum-based chemotherapy (carboplatin and pemetrexed) with a good response and progression-free survival (PFS) of 27 months. Following relapse, a primary culture, PA022, that was generated from malignant ascites confirmed dysfunctional HRR with concurrent sensitivity to PARPi on an ex vivo growth inhibition assay. Although there was disease progression following a re-challenge of chemotherapy, the patient was alive at the last follow-up review at 50 months. Figure 5B represents a patient diagnosed with non-epithelial ovarian cancer who underwent optimal cytoreductive surgery followed by 3 cycles of platinum-based chemotherapy (cisplatin and etoposide). A primary culture, PA024, that was generated from malignant ascites revealed competent HRR and resistance to PARPi ex vivo. Unfortunately, the patient quickly progressed and died within 7 months of diagnosis.

## 3. Discussion

The functional characterisation of patient primary cultures for HRR status is shown to be predictive of sensitivity ex vivo to PARPi and clinically to platinum sensitivity in ovarian cancer [8,14]. In this translational study, we demonstrate the ability to generate primary cultures from malignant ascites from patients with a variety of primary and metastatic cancers including GI, HB, pancreatic, and mesothelioma cancers. Cultures were grown as monolayers in a cobblestone pattern that is typical of epithelial cells, with further confirmation of their epithelial origin by pancytokeratin staining. Additionally, we show that it is feasible to undertake functional assays with the quantification of nuclear RAD51 foci to stratify cultures into HRC and HRD subgroups. Furthermore, we build on previous findings observed in ovarian cancer [14], demonstrating that HRD is associated with increased rucaparib sensitivity, at least in fresh cultures, supporting the potential application of PARPi in a wider group of malignancies. 

We have shown that HRR dysfunction is not restricted to ovarian cancer, with HRD detected in a proportion of all of the cancer types sampled. Recently, 102 HRR-related genes were tested in 8178 cancers from TCGA, demonstrating monoallelic and bialleleic mutations in only 13% and 5%, respectively [38]. Our experience of functional analysis in ovarian cancer suggests that HRD is seen in approximately 55% [8] of ovarian tumours, which is slightly more than the 51% estimate by TCGA [9]. Modest differences between functional analysis and genomic/epigenomic screening may be a reflection of post-translational inactivation of certain key enzymes and/or unknown factors (e.g. loss of 53BP1 or DNA-PKcs restores HRR in BRCA-mutated cells) [39]. The epigenetic silencing of BRCA1 accounted for 11% of the HRR defects reported in the TCGA ovarian cancer analysis [9], but was not investigated by Riaz et al. [38], who undertook whole-exome sequencing rather than the more sensitive whole genome analysis, offering further potential explanations for the apparent lower incidence of HRD in this dataset. Although our current study only includes a relatively small number of patient samples and more restricted tumour types, data from this study suggests a higher prevalence of HRD than the mutational analysis.

When presenting at late stage or relapse, the prognosis of the cancer types included in this study is poor, with second-line treatment options of limited efficacy, further emphasising the urgent need for additional therapies. Improved survival in even a small proportion of these patients through a stratified approach to the provision of platinum-based chemotherapy or PARPi therapy would potentially mark significant advancement in these aggressive tumour types. There is also a desire to test the efficacy of PARPi in patients without germline BRCA mutations, with several studies in ovarian cancer aiming to further define this subpopulation [40]. In fact, a clinical response to platinum (with PFS of more than 6 months) in ovarian cancer is itself a predictor of response to PARPi [41,42], and the basis of the current indication for maintenance therapy with all 3 licenced PARPi, so it may be possible to stratify at least a proportion of patients with other tumour types for PARPi therapy based upon their clinical response to previous lines of chemotherapy. PARPi are currently in clinical trials in pancreatic cancer patients with BRCA mutations [43,44,45], but there are also over 200 ongoing trials of PARPi in various other cancer types, which are not restricted to BRCA mutated cancer; some have had promising results [46,47]. Our previous studies in malignant pleural effusions [15] and the pilot data presented here indicate that HRR dysfunction is prevalent in a broad range of cancers, such that limiting trials to patients/tumours with germline or somatic BRCA mutations is likely to underrepresent the number who may benefit from PARPi therapy.

The primary goal of this study was to determine the feasibility of generating cultures from a variety of tumour types in which it was possible to assess HRR function to allow us to estimate the frequency of this defect. Patients with abdominal ascites represent a particular therapeutic challenge, as the development of ascites is usually indicative of a large volume of abdominal disease. Such patients frequently have a poor performance status, limiting their ability to tolerate systemic therapies. Several patients with advanced malignancy that were deemed unfit for standard of care chemotherapies were found to be HRD. Consideration of PARPi, which is known to have a lower toxicity and better tolerability in comparison to traditional chemotherapies, is perhaps of particular importance in this subgroup. 

Patient samples were collected at variable time points during their cancer treatment. HRD was observed in samples collected both before and after chemotherapy, although we appreciate that tumour biology may evolve with time and response to treatment. Typically, the provision of therapy is based on diagnostic and archival tumour histology, but as more targeted agents become available, it may be more beneficial to consider treatment according to the analysis of their molecular pathology/phenotype, including functional analyses in fresh samples, rather than cancer type alone. Previously, we have shown that HRR dysfunction is associated with improved 12-month survival following platinum-based chemotherapy in ovarian cancer [14]. Analysis of the present study suggests a modest survival benefit of 3 months in HRD patients (11 months) compared to HRC patients (8 months), as shown in Figure 4A. Improved survival is marginally greater in the subgroup of patients treated with platinum therapy, with an additional survival benefit of 5 months, as shown in Figure 4B. However, these observations should be interpreted with caution, given the limited sample size and the heterogeneous nature of the cohort. 

Optimising a clinically applicable assay to identify the HRD phenotype, which eliminates the need for lengthy cell culture, is still under investigation, with several commercial and research assays under ongoing development [5,6,10]. All of these assays have their strengths and limitations (reviewed in [48]). Despite the technical challenges associated with the RAD51 foci assay, it does demonstrate HRR function, independent of mechanism, in real time. With the ability to maintain viable cultures under conditions mimicking shipment to a centralised laboratory, for some tumours at least, this approach is feasible [49]. Recent studies using the ability to form RAD51 foci in ovarian cancer organoids confirms that this assay is predictive of response to PARPi [50].

Likewise, the importance of applying appropriate ex vivo methods to test clinical parameters is also highlighted in our rucaparib sensitivity data where there was a difference in response according to HRR status when using continuous 10-day rucaparib exposure, but not following 48 h exposure, as shown in Figure 3B. This is consistent with clinical trial data, which reports that a continuous dosing schedule of rucaparib is required for clinical response [35], and supports the methodological amendment to use continuous drug exposure for future experiments.

Replicating a high quality in vitro model of tumour state with the use of primary cell culture is also a critical factor in assay performance, but technical difficulties, not only in initially establishing cultures, but maintaining them to survive adequate passages before they senesce, is a major obstacle. Re-establishing PA cultures from cryopreserved cell material was one way to overcome this. Although we report a difference in response to rucaparib treatment between HRD and HRC samples for all of the continuous exposure experiments, this is significantly enhanced when the three re-established PA cultures are excluded from analysis, as shown in Figure 3B. More cultures would need to be analysed fresh and following cryopreservation to determine the impact of freezing on the growth characteristics and hence response to PARPi.

Questions also remain regarding the optimal method for sampling patients’ tumours. We have previously demonstrated that tumour heterogeneity extends to HRR status in ovarian cancer [51], with variable HRR functionality in solid tumours sampled from different anatomical regions within individual patients. The primary cultures of ascitic cells, unlike diagnostic biopsy samples from a single area of a tumour, are likely to be derived from the most currently active areas of the tumour and hence give a “real time” evaluation immediately prior to treatment. Therefore, we believe that the identification of HRR status in ascitic cultures is likely to be predictive of response to therapy with PARPi and/or platinum. 

## 4. Materials and Methods

### 4.1. Chemicals and Reagents

All of the chemicals and reagents were obtained from Sigma Aldrich (Poole, UK) unless otherwise stated. The PARP inhibitor (formerly known as CO-338, AG-014699, and PF-01367338; Clovis Oncology), rucaparib, was a kind gift from Zdenek Hostomsky, Pfizer GRD (La Jolla, CA, USA).

### 4.2. Sample Collection and Generation of Primary Cultures

Ethical approval and written consent were obtained from North East Newcastle and North Tyneside 1 Research Ethics Committee for the collection of clinical material and patient data (REC 12/NW/0202, REC 12/NE/0395). The study was performed in accordance with the Declaration of Helsinki. 

Samples of malignant ascites were opportunistically collected from patients either undergoing surgery or the palliative drainage of ascites at the Queen Elizabeth Hospital, Gateshead, UK or the Northern Centre for Cancer Care, Freeman Hospital, Newcastle, UK; these samples were assigned PA (primary ascites) reference numbers. Selection criteria were broad in this feasibility study, and included ages >18 years, suspected or confirmed malignancy, clinically evident ascites, and the ability to give written informed consent. 

Briefly, ascites was aspirated into sterile containers, transported to the lab, and processed within 24 hours of harvest in compliance with UN3373 regulations for Category B biological substances. Patients were link anonymised with a unique identification code, and a clinical dataset was recorded. Samples were registered and handled according to the Human Tissue Act (2004) and local guidelines.

#### Primary culture

First, 20 mL of ascites was added to 20 mL of RPMI-1640 supplemented with 20% foetal calf serum and 1% penicillin / streptomycin, pre-warmed to 37 °C, in 75 cm^3^ culture flasks. Cultures were incubated at 37 °C, 5% CO_2_, 95% humidified air. Medium was replenished every 4 to 7 days until cultures reached 70% to 80% confluence. Cells were then passaged for continuous culture. Morphological features were studied under an Olympus CK40 inverted microscope at 206 magnification, and images were captured using VisiCam software (VWR International, Radnor, USA). All of the experiments were carried out on early passage cultures (<5). 

### 4.3. Immunofluorescent Assays 

Cells were seeded onto coverslips at a density of 0.5 × 10^5^ cells/mL, and after 24 to 48 h, they were fixed and permeabilised with ice-cold methanol.

#### 4.3.1. Epithelial Characterisation

As in our previous studies [8,49], epithelial cell origin was confirmed by the examination of cell morphology, the immunoflourescent staining of fixed cells for pancytokeratin, as well as standard diagnostic cytology and immunohistochemistry examination. 

Ascitic culture cells were seeded onto coverslips, fixed, and hydrated. Coverslips were blocked with potassium chloride (KCl) buffer (120 mmol/L of KCl, 20 mmol/L of NaCl, 10 mmol/L of Tris-HCl, 1 mmol/L of ethylenediaminetetraacetic acid EDTA plus 0.1% Triton X-100) containing 2% (*w/v*) bovine serum albumin (BSA), and then incubated with 1:100 fluorescein isothiocyanate FITC-conjugated antipancytokeratin antibody (Merck Millipore, Watford, UK) for 1 h. Coverslips were mounted onto slides with 4´,6-diamidino-2-phenylindole (DAPI) mounting medium (Vectashield, Peterborough, UK). Cultures were examined with a Leica DMR fluorescent microscope (Leica microsystems GmbH, Wetzlar, Germany) and considered epithelial in origin if they contained >95% cytokeratin-positive cells.

#### 4.3.2. RAD51 Focus Assay for Assessment of HRR Function

We developed an assay of HRR function in a panel of cell lines with known HRR status, where a ≥twofold induction of RAD51 foci as well as γH2AX foci was found to correlate with HRR function [37]. We have used this assay to determine the HRR status of primary cultures of malignant pleural effusion cells [15] and ovarian cancer ascites cells, where it was predictive of ex vivo PARPi sensitivity [8] and response to carboplatin clinically [14]. Using this assay, primary cultures were evaluated for HRR function by the ability to form RAD51 foci in response to replication fork collapse. Replication fork collapse, as identified by γH2AX focus formation, was induced with 10 µM of rucaparib (a potent PARPi) for 24 to 48 h before fixation and permeabilisation, as previously described [8,14,51]. Coverslips were blocked with KCl buffer containing 2% BSA (*w/v*), 10% milk powder (*w/v*), and 10% goat serum (*v/v*), and then incubated with 1:100 anti-RAD51 antibody (PC130 Calbiochem, Merck Millipore) overnight at 4 °C. Coverslips were washed in buffer and incubated with 1:1000 anti-phospho-histone γH2AX (Ser139) (clone JBW301, Merck Millipore). Following further washing, coverslips were incubated in darkness with Alexa Fluor 546 Goat anti-mouse and Alexa Fluor 488 Goat anti-rabbit secondary antibodies (Invitrogen, Life Technologies, Paisley, UK), 1:1000 for 1 h. Coverslips were mounted on slides with DAPI. γH2AX and RAD51 foci were imaged using the Leica immunofluorescent microscope and quantified in at least 100 nuclei from different fields per sample using the ImageJ software, as previously described [49] in both treated (DNA damage-induced) and control cells. It has long been known that when PARP is inhibited, endogenous DNA single-strand breaks (SSBs) accumulate, and that when the replication fork encounters a SSB, a collapsed replication fork that is associated with a single-ended DNA DSB will result, triggering the phosphorylation of H2AX to give γH2AX foci. Therefore, PARPi-induced γH2AX foci will only form during S-phase. The accumulation of γH2AX foci was used as a pharmacodynamic biomarker of PARP inhibition in one of the earliest clinical trials [52]. Hair follicles were used in this trial, because the γH2AX foci only form during S-phase, and so circulating lymphocytes could not be used, as they are non-replicating. The mean number of foci in treated cells was divided by the mean number of foci in control cells to calculate the fold induction of γH2AX and RAD51 for each sample. A >2-fold increase in γH2AX was taken as an indication of sufficient replication fork collapse to induce HRR, whilst a <t2-fold induction was taken to indicate a failure to induce collapsed replication forks. In those cultures with a >2-fold increase in γH2AX, a >2-fold increase in RAD51 foci indicates HRR competence (HRC). Failure to increase RAD51 foci, despite an increase in γH2AX foci, was taken as indicative of an HRR defect (HRD).

### 4.4. Inhibition of Growth by Rucaparib

Primary cultures were seeded into 96-well plates at 1000 cells/well and allowed to attach for 24 h before treating with increasing concentrations of rucaparib in medium with 0.5% DMSO. Once cultures were established, they were continuously exposed to 0 to 100 µM for 10 days. Samples PA006, PA007, PA009, PA010, PA012, PA014, PA017, PA021, and PA022 were re-established from cryopreserved cells that had been frozen at the first or second passage of culture, and were treated with 0 to 30 µM of rucaparib for 48 h before replacing it with drug-free medium and incubated until untreated controls were just subconfluent. Three PA samples, PA006, PA007, and PA010, underwent both 48 h and continuous 10-day exposure treatments in parallel to the compared methodology. Cells were fixed with methanol:glacial acetic acid (3:1 *v/v*) and stained with 0.4% sulforhodamine B (SRB), as previously described [53]. Absorbance, relative to blank wells, was quantified by spectrophotometry on an Omega FLUOstar plate reader (BMG Labtech Ltd., Aylesbury, UK) using a 570-nm filter. The increase in cell density (SRB absorbance) of treated cells was calculated as a percentage of untreated controls (GraphPad Software, Inc., San Diego, CA).

### 4.5. Clinical Response and Survival Data

Patient survival data were calculated using the date of diagnosis, which was defined as the date of the histological confirmation of malignancy. Response to chemotherapy was determined by serial clinical examinations and computed tomographic (CT) scans at predefined intervals, including the completion of chemotherapy, unless new onset symptoms warranted urgent evaluation. Comparison was made with pre-treatment imaging. Response to treatment was determined by the treating oncologist using radiologic evidence of complete (CR) or partial response (PR). Progressive (PD) or recurrent disease (RD) was defined as the presence of an increasing volume of measurable disease on the CT scan, increasing serum tumour markers (where applicable), or clinical symptoms. Disease was defined as platinum-sensitive when there was no tumour progression within six months of completion of chemotherapy, and platinum-resistant when there was evidence of tumour progression within six months. For overall survival (OS), patients who died at follow-up (any cause) were uncensored. Patients alive at follow-up were censored [54].

## 5. Conclusions

In conclusion, our studies reveal that HRR status can be determined in abdominal ascites from a variety of tumour types, as presented here, or pleural effusions, as described previously [15]. In this study, around a third of tumours presenting with abdominal ascites were HRD, including those not generally associated with BRCA mutations. These data suggest that further investigations are warranted to determine the incidence of HRD in cancers of GI, pancreatic, HB, and mesothelioma origin. Furthermore, we believe that the potential of PARPi therapy in these tumour types is worthy of exploration.

## Figures and Tables

**Figure 1 cancers-11-00354-f001:**
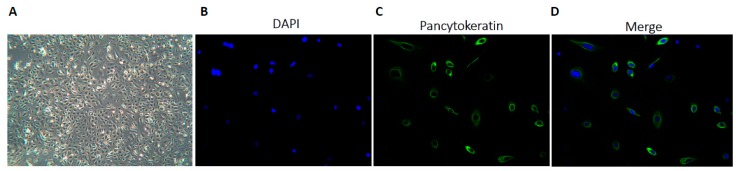
Epithelial characterisation: (**A**) Brightfield microscopy of primary culture established from PA014 with pancreatic cancer. Representative image demonstrates an adherent cobblestone monolayer morphology at 100× magnification. 10^5^ cells were seeded onto sterilised coverslips and after adherence, fixed and permeabilised with methanol. 4´,6-diamidino-2-phenylindole (DAPI) nuclear stain demonstrates viable cells (**B**) With pancytokeratin demonstrated in green, and (**C**) Following incubation with fluorescein isothiocyanate FITC-conjugated antipancytokeratin antibody at 1:100 for 1 hour. (**D**) Merged images. B, C and D were taken at 400× magnification.

**Figure 2 cancers-11-00354-f002:**
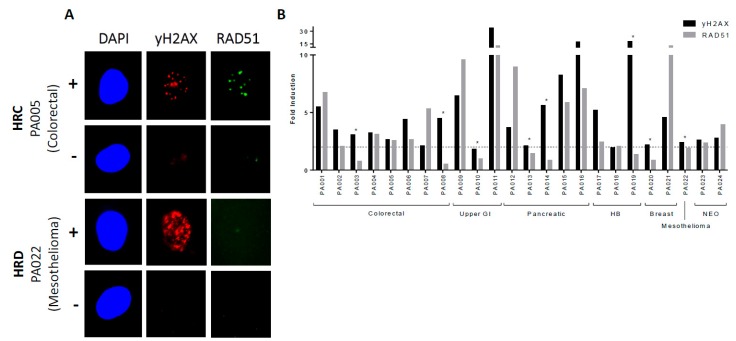
Determination of HRR status. (**A**) Immunofluorescence microscopy of RAD51 foci assay in malignant cells treated with rucaparib (+) and in untreated controls (–) for PA005 and PA022 cultures. Cell nuclei were stained with DAPI (blue), before quantifying collapsed replication forks (indicated by γH2AX foci (red) and DNA repair by HRR by RAD51 foci (green) using ImageJ software. Samples were characterised as HRR competent (HRC, PA005) if a >2-fold increase in RAD51 was observed in comparison to untreated controls following the induction collapsed replication forks. HRR-deficient (HRD, PA022) cultures have <2-fold increase in RAD51 foci. Magnification: 400×. (**B**) HRR status of primary cultures categorised by tumour type. Mean fold increase in γH2AX and RAD51 foci per cell were calculated by ImageJ analysis and plotted using GraphPad Prism 6 software. The 2-fold induction threshold is represented by the dotted line. Although in PA010, rucaparib only increased γH2AX by 1.84-fold, RAD51 foci were not increased at all. The exposure of these cells to 5 mM of hydroxurea induced collapsed replication forks, and caused a nearly 8-fold increase in γH2AX, but only a 1.8-fold increase in RAD51. Following 2 Gy irradiation, there was a 22-fold increase in H2AX, but no increase in RAD51 (data not shown). Therefore, we designated this primary culture as being HRD *denoted HRD cultures. GI = gastrointestinal, HB = hepatobiliary, NEO = non-epithelial ovarian.

**Figure 3 cancers-11-00354-f003:**
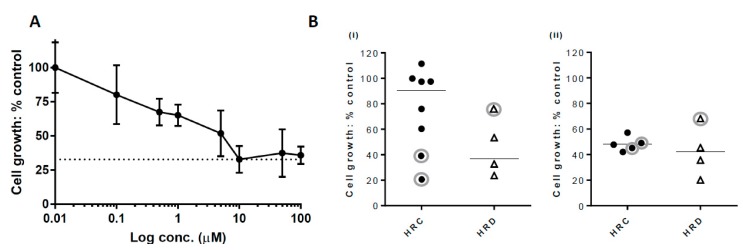
Growth inhibition by rucaparib treatment (**A**) Growth inhibition curve of PA008 culture (HRD) with rucaparib. Cell proliferation was assessed by sulforhodamine B assay. Data are the mean values of six repeats for each concentration, with error bars demonstrating SD. Cell growth, relative to control following exposure to 10 µM of rucaparib, was 32.8% (dotted line). (**B**) Cell growth distribution by HRR status after treatment with 10 µM of rucaparib. Values were calculated as growth following rucaparib treatment as a percentage of DMSO only control by sulforhodamine B (SRB). Data are the mean values of six repeats. (**i**) % cell growth: control of all cultures treated continuously for 10 days. Non-circled data points are fresh cultures, circled data points are cultures that were re-established from thawed cryopreserved cells. Mean values, represented by horizontal bars are for fresh cultures only (excluding three circled data points) (**ii**); % cell growth: control of thawed cultures treated for 48 h, circled data points represent the samples that were also treated continuously in (i).

**Figure 4 cancers-11-00354-f004:**
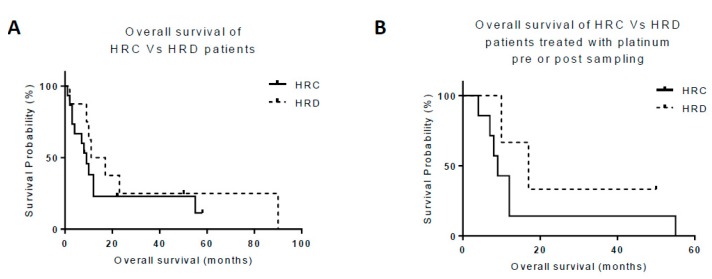
Patient survival. Kaplan–Meier survival curves for overall survival (OS) in HRR-competent (HRC) versus HRR-deficient (HRD) cancers. (**A**) Median OS in patients with HRC cultures was 8 months (*n* = 12) in comparison to 11 months in all of the patients with HRD cultures (*n* = 8). (**B**) In patients treated with platinum-based chemotherapy, the median OS in patients with HRC cultures was nine months (*n* = 7) in comparison to 17 months in those with HRD cultures (*n* = 3).

**Figure 5 cancers-11-00354-f005:**
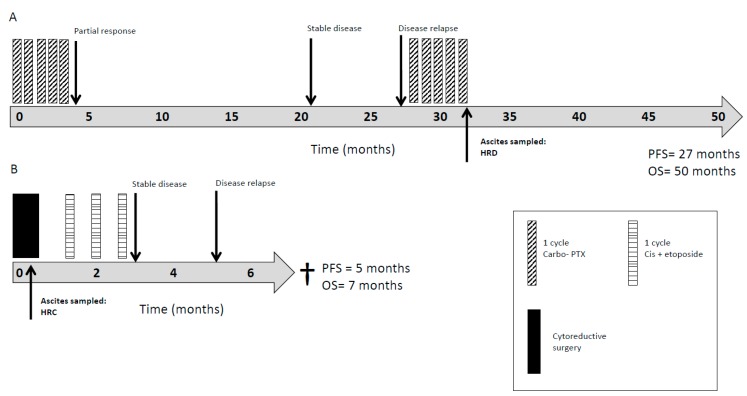
Patient survival in HRD vs. HRC. Clinical course of (**A**) HRD patient PA022 (peritoneal mesothelioma). Patient alive at last review. and (**B**) HRC patient PA024 (non-epithelial ovarian) Progression-free survival (PFS) was calculated as the time (months) elapsed from the date of diagnosis to the date of the first relapse. Overall survival (OS) was calculated as the time (months) elapsed from the date of diagnosis to the date of death.

**Table 1 cancers-11-00354-t001:** Clinical data for all 24 established cultures. Patient characteristics, ascites sample details and pre-sampling and post-sampling treatment data. Homologous recombination DNA repair (HRR)-competent samples are denoted as C, and HRR-deficient samples are denoted as D. Where cytotoxicity to poly(ADP-ribose) polymerase inhibitors (PARPi) was not determined, samples are marked with an X. Overall survival (OS) was calculated as the time (months) elapsed from the date of diagnosis to the date of death, OS figures denoted with ‘ are censored patients that were alive at last review. ***This patient received panitumumab as part of an overseas trial.

Cancer Type	Age (years) Median (Range)	Proportion Male n (%)	Proportion Femalen (%)	Sample ID	Sampling Imepoint (Pre/post treatment)	% Pancytokeratin Positive Cells	HRR Status	Cell Growth with 10 µM Rucaparib (% Control)	Stage at Diagnosis (TNM / FIGO)	Pre-sample Treatment	Post-sample Treatment	Overall Survival (Months)
COLORECTAL n = 8 (33%)	55 (43–83)	1 (4%)	7 (29%)	PA001	Pre	100	C	X	T4bNXM1	Surgery	FOLFOX	8
				PA002	N/A	95	C	X	T4bN1M1	Surgery	None	2
				PA003	Pre	100	D	23.6	T4bN1M1	Surgery	FOLFOX	17
				PA004	Pre	94	C	111.5	T4bNXM1	Surgery	FOLFIRI + bevacizumab	10
				PA005	N/A	100	C	60.3	T4bNXM1	None	None	1
				PA006	Post	93	C	39	T4NXM1	1. FOLFOX + bevacizumab 2. FOLFIRI + cetuximab	Radiotherapy	12
				PA007	Post	98	C	20.7	T3N2M0	Neoadjuvant chemoradiotherapy (capecitabine); Surgery; Adjuvant CapOX; Following metastatic diagnosis: 1. Capecitabine, bevacizumab; 2. FOLFOX; 3. Irinotecan, cetuximab	4. Phase 1 trial (gemcitabine + VX-970 ATRi)	55
				PA008	Pre	96	D	32.8	T4NXM1	Surgery	FOLFIRI + bevacizumab	23
UPPER GI n = 3 (13%)	52 (51–72)	0	3 (13%)	PA009	Post	93	C	42	T4NXM1	Capecitabine	None	22’
				PA010	Post	97	D	76.1	T4NXM1	EOX	None	10
				PA011	Pre	93	C	75.9	T4bNXM1	Surgery	Radiotherapy	4
PANCREATIC n = 5 (21%)	49 (48–77)	2 (8%)	3 (13%)	PA012	Post	96	D	49.7	T4NXM1	Radiotherapy	None	11
				PA013	N/A	99	C	97.4	TXNXM1	None	None	3
				PA014	N/A	97	D	45.5	T4NXM1	None	None	2
				PA015	Post	96	C	X	T4NXM1	Gemcitabine	Radiotherapy	8’
				PA016	Post	99	C	X	T4NXM1	FOLFIRINOX	None	4
HB n = 3 (13%)	66 (44–71)	3 (13%)	0	PA017	Pre	97	C	57.1	T3NXM1	Cisplatin/gemcitabine + panitumumab***	None	9
				PA018	Post	95	C	X	T4NXM1	Cisplatin/gemcitabine	None	12
				PA019	Post	98	D	X	T4NXM1	Phase 1 trial (dexanabinol, sorafenib)	None	9
BREAST n = 2 (8%)	67 (66–68)	0	2 (8%)	PA020	Post	100	D	53.4	T2N2M0	Surgery; Adjuvant radiotherapy; Adjuvant anastrazole	1. Letrozole 2. Capecitabine + vinorelbine 3. Exemestane + everolimus	90
				PA021	Post	99	C	47.8	UN	Paclitaxel	UN	UN
MESOTHELIOMA n = 1 (4%)	65	0	1 (4%)	PA022	Post	97	D	35.6	Primary peritoneal mesothelioma (epitheloid subtype)	Carboplatin + pemetrexed	Rechallenge carboplatin + pemetrexed	50’
NEO n = 2 (8%)	44 (26–63)	0	2 (8%)	PA023	Pre	93	C	99.9	1C	Surgery	BEP	58’
				PA024	Pre	98	C	97.4	3C	Surgery	Cisplatin + etoposide	7
All n = 24	57 (26–83)	6 (25%)	18 (75%)									

FOLFOX = 5-fluorouracil, folinic acid, oxaliplatin; FOLFIRI = 5-fluorouracil, folinic acid, irinotecan; CapOX = Capecitabine, oxaliplatin; EOX = Epirubicin, oxaliplatin, capecitabine; FOLFIRINOX = 5-fluorouracil, folinic acid, irinotecan, oxaliplatin; BEP = Bleomycin, etoposide, cisplatin; UN = UNKNOWN.

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
