# Peer review of "Exploring the Frequency of Homologous Recombination DNA Repair Dysfunction in Multiple Cancer Types"

_cancers, 2019, doi:10.3390/cancers11030354_

Round 1
Reviewer 1 Report
It is difficult to make robust conclusions with the data presented. Would recommend a larger cohort of patients in each category.
RAD51 focus assay for assessment of HRR (dys)function may not be adequate to establish whether a tumor has an inherent defect in the HR DDR pathway defect(s).
Have the authors performed NGS on primary tumor vs. ascitic fluid tumor samples to establish HR DDR pathway aberrations? This information may help corroborate results from the Rad51 focus assay.
Fig 4. Kaplan-Meier survival curves are difficult to interpret with such small numbers of patients.
Figure 1 legend - 5 x 10x x=?
Author Response
1. It is difficult to make robust conclusions with the data presented. Would recommend a larger cohort of patients in each category.
We agree with the reviewer that a robust analysis of a small sample but as we acknowledge in the manuscript, and have further clarified in the text (abstract) this was a feasibility study as we had stated in the Introduction (last paragraph) and Discussion (paragraph starting line 204 clean, line 213 tracked changes). In the Discussion we had stated that our “observations should be interpreted with caution given the limited sample size and the heterogeneous nature of the cohort. The primary goal of this study was to determine the feasibility of generating cultures from a variety of tumour types in which it was possible to assess HRR function to allow us to estimate the frequency of this defect.” Indeed much of our Discussion does acknowledge the limitations of the sample size (e.g. lines 185-7 clean, lines 186-188 tracked changes) and lessons we have learnt – e.g. the need for continuous exposure in growth inhibition assays. Nevertheless to reinforce this message we have modified the Discussion (lines 200-1 clean, 202-3 tracked changes). With regard to selecting a larger cohort in a particular category our aim was to determine the feasibility of evaluating HRR status in a variety of tumour types and to get an overall impression of the frequency of defects, rather than identify the frequency in a specific cancer type. We believe that the future of targeted therapy will require a move towards selecting patients on the basis of the molecular pathology/phenotype of their tumour, rather than its tissue of origin. This message has previously been promoted (e.g. ref 1) and we clarify further in the Discussion (line 216-8 clean, 226-9 tracked changes) so increasing the cohort of patients in a particular category is not necessarily relevant. HRR status is likely to be a predictive biomarker as HRD confers sensitivity to DNA damaging chemotherapy, particularly cis/carboplatin, which is why we specifically looked at patients receiving platinum therapy.
2. RAD51 focus assay for assessment of HRR (dys)function may not be adequate to establish whether a tumor has an inherent defect in the HR DDR pathway defect(s). Have the authors performed NGS on primary tumor vs. ascitic fluid tumor samples to establish HR DDR pathway aberrations? This information may help corroborate results from the Rad51 focus assay
As we indicate in the Discussion, there are strengths and weaknesses to all assays for HRR function, NGS, aCGH, LOH etc. and these have recently been reviewed (reference 46) but see also Hill SJ, et al Prediction of DNA Repair Inhibitor Response in Short-Term Patient-Derived Ovarian Cancer Organoids Cancer Discov. 2018;8:1404-1421 https://www.ncbi.nlm.nih.gov/pubmed/30213835 (now inserted reference 50), which describes the use of a similar RAD51 focus assay in patient-derived organoids to assess HRR function and indeed the authors comment on the clinical utility of this assay. Our first study in ovarian cancer was the first indication that ≥50% of HGSOC were HRD, before the TCGA analysis revealed a similar figure (refs 8 and 9) and was predictive of response (ref 14). Our previous experience with NGS in malignant pleural effusions suggests that aberrations in DNA repair genes are widespread in cancer, independently of HRR function (ref 15) and we have previously investigated the heterogeneity of solid tumour deposits, including with respect to HRR status (ref 51) but we hypothesise that the ascites are derived from the most active deposits and therefore potentially most predictive of response.
Fig 4. Kaplan-Meier survival curves are difficult to interpret with such small numbers of patients.
We agree with the reviewer that the ability of a Kaplan-Meier to provide an accurate probability of survival is dependent upon the number of observations included in each cohort. Thus the curves included in the manuscript should be interpreted with caution, as we have acknowledged (in line 223-5 clean, 233-4 tracked changes). Despite the small number of patients included, the curves do clearly demonstrate that the patients treated for advanced and recurrent cancer in this study have overall very poor survival, further highlighting the urgent need for exploration of novel agents in this setting.
The suggestion of a small improvement in overall survival of tumours categorised as HRD in comparison to HRC, might suggest favourable tumour biology and the subgroup analysis seen in Figure 4B suggests preferential response to platinum in HRD tumours. Since it is now universally accepted in clinical ovarian cancer studies that response to platinum is predictive of response to PARPi (see also changes to text line 193-7 clean 195-9 tracked changes and additional references 41 and 41), we can hypothesize that HRD is a biomarker of response to platinum and PARPi reinforcing the potential benefit of PARPi in patients with various tumour types. Despite the small numbers included the cautious analysis of the limited survival data does at least suggest favourable prognosis and response to platinum (with hypothesised response to PARPi) in HRD tumours supporting further studies in non-ovarian tumour types.
Figure 1 legend - 5 x 10x x=?
This error has been corrected, it should have been 105
Reviewer 2 Report
The authors already established an assay for HRR function in ovarian cancer ascites predicting ex vivo sensitivity to PARPi and clinical sensitivity to platinum therapy, successfully applied to malignant pleural effusions. By the evidence suggesting that HRR may be important in many other cancers, the novelty of this study is to determine the feasibility of assessing HRR function and PARPi sensitivity in ex vivo primary cultures from patients’ malignant ascites from a variety of cancer types.
This represents still an unexplored area with interesting perspectives for a possible broader use of PARPi.
Moreover the prognostic value of this analysis is highlighted, together with its usefulness in determining the efficacy of a therapy.
Methodology is rigorous, detailed and clearly described, from characterization of ascite primary cultures, determination of the frequency of HRR dysfunction (HRD), to the assessment of HRR function and clinical trials/survival data.
Statistical analysis is accurate. The use of the non-parametric Mann-Whitney test, allows a correct evaluation of data from the examined populations.
I reccomend publication in Cancers in its present form.
Author Response
We thank the reviewer for this positive view
Reviewer 3 Report
The authors describe a first attempt to analyze a diverse array of cancer samples for Homologous Recombination activity. They developed a primary cell culture system for dissociated tumor samples and investigate the reaction to treatment with the PARP inhibitor Rucaparib. They found some differential reactions, but most analyses lacked sufficient statistical power to draw any definitive conclusions.
This type of characterization is worthwhile, but it is a pity that the group of tumors was very diverse and did not allow any conclusions to be drawn. There are also some technical issues that need to be resolved before further studies should be undertaken, as outlined below.
The major point of attention is the definition of Homologous Recombination Deficiency (HRD). The authors define this as samples in which the induction of cells carrying RAD51 foci has less than doubled upon Rucaparib treatment and where the number of cells carrying gamma-H2AX foci has more than doubled. As can be observed in figure 2B, this results in several examples where the gamma-H2AX has just over doubled and RAD51 just less than doubled (PA013 and PA025). Both are called HRD in the paper, but what is the rationale for this? PA010 is also called HRD, whereas gamma-H2AX less than doubled.
This type of classification seems highly debatable. The main trouble is, that less than doubled RAD51 foci might be the result of low proliferation, as double strand break formation is dependent on replication. Furthermore, cells need to be in the S or G2 phase of the cell cycle, because otherwise RAD51 foci are not formed. Therefore, the authors should show that there are indeed sufficient cells in S/G2 phase and preferentially do a double staining for cell cycle phase and RAD51 foci. Without such analysis, the inappropriate label HRD may be attached to tumors that are really HRP.
Furthermore, it would be highly informative to do the same RAD51 foci analysis after ionizing radiation, which does not depend on replication to induce the DNA breaks. This might even reveal tumors that lack RAD51 foci upon PARP inhibitor, but able to form these foci after irradiation (cause by e.g. PARP1 deficiency). Such tumors are not HRD and might even be highly PARP inhibitor resistant, whereas true HRD tumors should be sensitive. In the absence of additional analysis, it is impossible to draw conclusions about the HRD nature of the tumors.
Minor points:
Figure 1 shows a field of cells in which not all cells stain positive for cytokeratin. Could the authors give a clear quantification of the percentage of cytokeratin positive cells in each culture, to exclude that data are influenced too much by contaminating non-epithelial cells?
What is the meaning of ‘greater than 2-fold induction of RAD51 foci’? Does this mean the number of nuclei with a minimum number of foci or does it mean the average number of foci per nucleus? According to the methods section the latter explanation is correct. However, it might be better to take the first one, as the average number of foci is heavily influenced by the proliferation rate. The authors should explain their choices. Even better, they should confine their analysis to S/G2 phase cells, as explained above.
Author Response
[Text given here, website would not let me upload word document with figure]
Reviewer 3
The authors describe a first attempt to analyze a diverse array of cancer samples for Homologous Recombination activity. They developed a primary cell culture system for dissociated tumor samples and investigate the reaction to treatment with the PARP inhibitor Rucaparib. They found some differential reactions, but most analyses lacked sufficient statistical power to draw any definitive conclusions.
1. This type of characterization is worthwhile, but it is a pity that the group of tumors was very diverse and did not allow any conclusions to be drawn.
We agree with the reviewer that a robust analysis of a small sample but as we acknowledge in the manuscript, and have further clarified in the text (abstract) this was a feasibility study as we had stated in the Introduction (last paragraph) and Discussion (paragraph starting line 204 clean, line 213 tracked changes). In the Discussion we had stated that our “observations should be interpreted with caution given the limited sample size and the heterogeneous nature of the cohort. The primary goal of this study was to determine the feasibility of generating cultures from a variety of tumour types in which it was possible to assess HRR function to allow us to estimate the frequency of this defect.” Indeed much of our Discussion does acknowledge the limitations of the sample size (e.g. lines 185-7 clean, lines 186-188 tracked changes) and lessons we have learnt – e.g. the need for continuous exposure in growth inhibition assays. Nevertheless to reinforce this message we have modified the Discussion (lines 200-1 clean, 202-3 tracked changes). With regard to selecting a larger cohort in a particular category our aim was to determine the feasibility of evaluating HRR status in a variety of tumour types and to get an overall impression of the frequency of defects, rather than identify the frequency in a specific cancer type. We believe that the future of targeted therapy will require a move towards selecting patients on the basis of the molecular pathology/phenotype of their tumour, rather than its tissue of origin. This message has previously been promoted (e.g. ref 1) and we clarify further in the Discussion (line 216-8 clean, 226-9 tracked changes) so increasing the cohort of patients in a particular category is not necessarily relevant. HRR status is likely to be a predictive biomarker as HRD confers sensitivity to DNA damaging chemotherapy, particularly cis/carboplatin, which is why we specifically looked at patients receiving platinum therapy.
1. There are also some technical issues that need to be resolved before further studies should be undertaken, as outlined below.
The major point of attention is the definition of Homologous Recombination Deficiency (HRD). The authors define this as samples in which the induction of cells carrying RAD51 foci has less than doubled upon Rucaparib treatment and where the number of cells carrying gamma-H2AX foci has more than doubled. As can be observed in figure 2B, this results in several examples where the gamma-H2AX has just over doubled and RAD51 just less than doubled (PA013 and PA025). Both are called HRD in the paper, but what is the rationale for this? PA010 is also called HRD, whereas gamma-H2AX less than doubled.
This type of classification seems highly debatable. The main trouble is, that less than doubled RAD51 foci might be the result of low proliferation, as double strand break formation is dependent on replication. Furthermore, cells need to be in the S or G2 phase of the cell cycle, because otherwise RAD51 foci are not formed. Therefore, the authors should show that there are indeed sufficient cells in S/G2 phase and preferentially do a double staining for cell cycle phase and RAD51 foci. Without such analysis, the inappropriate label HRD may be attached to tumors that are really HRP.
Furthermore, it would be highly informative to do the same RAD51 foci analysis after ionizing radiation, which does not depend on replication to induce the DNA breaks. This might even reveal tumors that lack RAD51 foci upon PARP inhibitor, but able to form these foci after irradiation (cause by e.g. PARP1 deficiency). Such tumors are not HRD and might even be highly PARP inhibitor resistant, whereas true HRD tumors should be sensitive. In the absence of additional analysis, it is impossible to draw conclusions about the HRD nature of the tumors.
We thank the reviewer for this careful consideration of the data, indeed it is something we have discussed ourselves at length and it is best to address all these points together by considering the mechanism of DNA damage induction and its repair.
When PARP is inhibited endogenous DNA single strand breaks (SSBs) will remain unrepaired and accumulate. When the replication fork encounters a SSB a collapsed replication fork, associated with a single-ended DNA DSB will result. This triggers the phosphorylation of H2AX (by ATM, ATR and possibly DNA-PK) to give gH2AX foci. Therefore PARPi-induced gH2AX foci will only form during S-phase and gH2AX foci can be taken as an indication of S-phase cells in the same way as geminin or cyclin E expression can. The Methods text has been modified to clarify the issue (Lines 319-322 clean, 329-332 tracked changes)
On the other hand, gH2AX foci will form in cells post-irradiation, whether they are HRR functional or dysfunctional (reference 37 Drew et al 2011 figure 2A – reproduced here). White bars are 2 Gy IR, black bars are 24 h 10 uM rucaparib (AG14699). note HCC1937 are paired BRCA mutant and corrected cells) and in all phases of the cell cycle, including G1/G0. Since RAD51 foci will only form in S/G2 phase cells the failure to form RAD51 foci after irradiation could result from a G1/G0 population or HRD and it is not possible to distinguish the two causes.
The difference in the height of the gH2AX bars after rucaparib exposure, in reference 37 was somewhat reflective of the cell doubling time and hence likely S-phase fraction, and we assume this is the case also for the current manuscript.
Whilst the 2-fold increase cut-off may appear somewhat arbitrary it is based on our experience in multiple cell lines and primary cultures of patient samples, both unpublished and published, and was determined by a statistical analysis of the data reported in ref 37 and used in our subsequent studies in ovarian cancer ascites and malignant pleural effusions. It is likely there is a spectrum of HRR defects that reflect mutations/ expression/activity of genes/proteins that play a greater or lesser role in the pathway, e.g. a BRCA2 mutation may have a greater effect on HRR than say one in CtIP or ATM.
We believe cells lacking PARP would not show an increase in gH2AX foci after exposure to rucaparib as such cells would have compromised SSB repair and there should therefore be a similar number of unrepaired SSBs in the control.
With regards to PA010, we agree with the reviewer’s acute observation that this failed to reach the threshold for gH2AX formation. We now include the data (in the figure legend) that following treatment with hydroxurea (to collapse replication forks) there was a nearly 8-fold increase in gH2AX but only 1.8-fold increase in RAD51. Following irradiation there was a 22-fold increase in H2AX but no increase in RAD51.This led us to conclude that this culture was HRD.
Minor points:
1. Figure 1 shows a field of cells in which not all cells stain positive for cytokeratin. Could the authors give a clear quantification of the percentage of cytokeratin positive cells in each culture, to exclude that data are influenced too much by contaminating non-epithelial cells?
The percentage of cells positive for cytokeratin are now given in table 1 (column 7)
2. What is the meaning of ‘greater than 2-fold induction of RAD51 foci’? Does this mean the number of nuclei with a minimum number of foci or does it mean the average number of foci per nucleus? According to the methods section the latter explanation is correct. However, it might be better to take the first one, as the average number of foci is heavily influenced by the proliferation rate. The authors should explain their choices. Even better, they should confine their analysis to S/G2 phase cells, as explained above.
The induction is calculated from the number of foci/cell for the entire population as we have done for all our previous publications using this assay. As explained above, the increase in gH2AX foci is taken to indicate S-phase cells.

Round 2
Reviewer 1 Report
All questions have been answered.
Author Response
We thank the reviewer for accepting our modifications
Reviewer 3 Report
The major point is the quantification of RAD51 foci and determination of the cut off point for Homologous Recombination Deficiency (HRD). The cut off is still rather arbitrary and without much evidence that this is the correct point. For example, no BRCA gene mutation or epigenetic silencing has been determined, so it is not possible to assess whether all BRCA tumors are really in the HRD class. Furthermore, it is not clear whether all HRD tumors according to this classification are really in the right category.
It would have been much more convincing if the authors could show for at least a subset of tumors that indeed this way of classification results in a similar scoring as a combination with a cell cycle marker (such as geminin). Theoretically, they are correct in the assumption that only in S phase the SSB give rise to DSB and gamma-H2AX and RAD51 foci. However, gamma-H2AX has been found after other types of DNA damage in several studies, so a positive proof of their assumption would be much better than assuming that this is the case.
Other points have been addressed in a satisfactory manner.
Author Response
In response to Reviewer 3’s comments, in addition to the amendments previously made we have now added further references (line 357-361) to our publication in J. Nat’l Cancer Inst [37] which describes how we derived the cut-off using a panel of HRR competent and dysfunctional cells. We also refer to our publications using this assay in primary cultures of malignant ovarian ascites cells, where it correlated with ex-vivo rucaparib sensitivity [8, Clinical Cancer Research] and clinically with survival [14, Cancer Research], and in malignant pleural effusions [15, Br J Cancer]. These are all high impact publications, having been cited 209, 206, 60 and 14 times, respectively. We believe that the most compelling validation of the assay is the clinical correlation with survival in patients with ovarian cancer treated with carboplatin [14], which is known to be more cytotoxic to HRR-defective cells.
We appreciate that γH2AX can be used to identify other DNA lesions, particularly DSBs, as well as stalled replication forks it is unlikely that rucaparib would induce such lesions directly. The increase in γH2AX foci following PARP inhibition is generally thought to be a result of an attempt to replicate unrepaired endogenous DNA SSB, which causes replication stress and stalled replication forks. Since we did not expose the cells to exogenous DNA damage and we determined the increase in γH2AX foci relative to control, we believe in this instance we are measuring replication stress. Indeed γH2AX foci have been proposed as a pharmacodynamic assay for PARP inhibition in replicating cells such as hair follicles (Redon et al 2010 Clinical Cancer Res https://www.ncbi.nlm.nih.gov/pmc/articles/PMC2940983/)
Whilst geminin can be used to identify cells in S and G2 phases of the cell cycle we believe it is more useful to identify cells that are undergoing replication stress and having stalled replication forks as a result of PARP inhibition, as it is these cells that need to engage HRR for their survival.